# Factors influencing farming households' climate change adaptation strategies in Central Vietnam

**Bui Huy Nhuong[1], Dinh Duc Truong[2]\*, Le Huy Huan[2], Bui Thi Hoang Lan[2], Nguyen Dieu Hang[2], Duong Duc Tam[2]**

**1** University Board of Management, National Economics University (NEU), Vietnam, **2** Faculty of Environmental, Climate Change and Urban Studies, NEU, Vietnam

\* truongdd@neu.edu.vn

## Abstract

This study investigates the determinants of household-level adaptation strategies to climate variability and saltwater intrusion in the coastal regions of Central Vietnam. Using a Multinomial Logistic Regression (MNL) model, the analysis is based on a cross-sectional survey of 356 farming households, focusing on four primary adaptation measures: vegetable production, shrimp farming, adoption of salt-tolerant rice varieties, and the lotus-fish farming model. The results reveal that socio-economic, demographic, environmental, and institutional factors significantly influence the adoption of specific adaptation strategies. Key determinants include gender, education, age, farming experience, household income, land characteristics, access to information, credit services, membership in civic organizations, and participation in training programs. Male-headed households and those with greater access to climate information and social networks were more likely to adopt diverse and complex adaptation strategies. In contrast, households with limited land resources, lower incomes, or lacking institutional support were less adaptive. The findings highlight the heterogeneity of adaptive behaviors and the need for tailored interventions. From a policy perspective, enhancing institutional capacity—especially through targeted training, increased access to subsidized credit, and support for community-based organizations—can significantly strengthen farmers' adaptive capacities. Moreover, the study contributes to filling key research gaps in the Southeast Asian context by integrating socio-economic and environmental variables into a unified analytical framework. These insights are critical for designing inclusive and effective climate adaptation policies aimed at safeguarding rural livelihoods and promoting sustainable agricultural development in climate-vulnerable regions.

**Data availability statement:** All data are in the manuscript and/or supporting information files.

**Funding:** This study is funded by the National Economics University, Vietnam. The funders had no role in study design, data collection and analysis, decision to publish, or preparation of the manuscript.

**Competing interests:** The authors have declared that no competing interests exist.

## 1. Introduction

Climate change and the increasing frequency of extreme weather events have had profound effects on agricultural systems, particularly in countries which geographical and climatic conditions render them highly sensitive to environmental variability [1–3]. These unpredictable climatic shifts not only undermine the quality and productivity of agricultural outputs but also compel farmers to invest more heavily in adaptation measures, ultimately reducing profit margins and exacerbating economic losses [4–7]. Alarmingly, approximately 52% of global agricultural land is currently classified as moderately to severely degraded [5]. The compounded effects of climate change and extreme weather can trigger a cascade of socio-economic instabilities, contributing to poverty and development challenges across household, community, and national scales [6–10].

For farmers, a key approach to mitigating these adverse impacts is to adjust agricultural systems in ways that enhance resilience to changes in temperature, precipitation patterns, and climate-induced hazards [11,12]. Broadly defined, climate change adaptation refers to adjustments in natural or human systems in response to actual or anticipated climatic stimuli or their effects, with the aim of moderating harm or exploiting potential benefits [13,14]. These behavioral adjustments can significantly reduce agricultural vulnerability and exposure to climatic risks [15,16]. Adaptation, however, is inherently complex, encompassing multi-dimensional and multi-scalar processes [17–19]. It may occur in various forms—autonomous or planned, and proactive or reactive—and can also be classified by temporal duration, spatial scope, and strategic approach [20,21]. Farmers' selection of adaptation strategies is typically influenced by a combination of demographic, socio-economic, institutional, and agro-ecological factors. Nonetheless, empirical evidence suggests that the effect of these determinants varies depending on the specific livelihood systems and agricultural contexts in which farmers operate [22,23]. Consequently, there is a pressing need to identify the key factors shaping adaptation choices and the capacity of farmers to mitigate climate change impacts on their environments, livelihoods, and associated socio-economic and institutional structures, thereby informing effective policy instruments and management frameworks [24–26].

The Asian region, including Vietnam, is among the most climate-vulnerable areas globally, characterized by rapidly shifting climatic conditions that pose substantial challenges for communities attempting to respond and adapt [27,28]. Vietnam, in particular, is identified as one of the five countries most severely affected by climate change, owing to its long coastline, high population density, and concentration of socio-economic activities in low-lying coastal areas [27,29,30]. According to Department of Climage Change (2020), a 100 cm rise in sea level could inundate approximately 40,000 km²—equivalent to 12.1% of the nation's current land area—potentially displacing up to 17.1 million people [27]. This rise would devastate the Red River and Mekong Deltas, two of the most critical agricultural and economic zones in the country [30]. Over the past decade, the economic damage attributable to climate change has been estimated at 2.5% to 3.0% of GDP annually, and in the absence of effective adaptation policies, projections indicate this figure may rise to

10% of GDP per year by 2050 [26,31]. The agricultural sector, which remains central to Vietnam's economy and food security, is particularly vulnerable. Climate change exerts adverse effects on agricultural production, threatening the livelihoods of farming communities, national food security, and the value of agricultural exports [32,33]. As one of the world's top five rice-exporting nations, Vietnam's agricultural supply chains are increasingly exposed to climate-induced risks [26,34]. Despite various initiatives, climate adaptation efforts in Vietnam continue to face multiple constraints, including high exposure to hazards, heightened sensitivity, and limited adaptive capacity, especially in resource-scarce rural areas [35–37].

Globally, a substantial body of literature has explored the determinants of climate change adaptation strategies among communities and agricultural households in both developed and developing economies [38–40]. These studies typically identify behavioral, psychological, and socio-economic characteristics as key influencing factors. Theoretical frameworks underlying such research often derive from behavioral models, technology acceptance theories, and collective action paradigms [41–43].

Despite the substantial progress in the global literature on climate change adaptation among farming households, important research gaps remain—particularly in the context of transitional economies such as Vietnam, where institutional reforms, environmental vulnerability, and livelihood diversification intersect in complex ways. First, while a significant body of empirical research has examined household-level adaptation in regions such as Sub-Saharan Africa and South Asia, there is a noticeable dearth of studies focusing on Southeast Asia, especially in Vietnam. Within this region, rural households residing in coastal and salinity-affected zones face unique climate risks, including saltwater intrusion, erratic rainfall, and seasonal droughts. Yet, household-level analyses of how these communities perceive, respond to, and cope with such hazards remain limited. As a result, current adaptation models may lack external validity when applied to Southeast Asian coastal contexts, where socio-cultural and agro-ecological dynamics differ substantially from those studied in African or South Asian cases. Second, the majority of existing research tends to analyze adaptation using either a single strategy (e.g., crop diversification or irrigation) or aggregate adaptation indices, without unpacking the distinct types of strategies households may pursue simultaneously. Such an approach overlooks the inherent heterogeneity, sequencing, and trade-offs embedded in farmers' adaptation decisions. In highly variable environments, households often adopt a combination of strategies based on resource constraints, perceived risks, and institutional access. Therefore, there is a critical need to disaggregate adaptation behavior across specific climate risks—such as drought versus saltwater intrusion—and to recognize that different strategies may be complementary, substitutive, or even conflicting. Third, although socio-economic, environmental, and institutional drivers of adaptation have been studied individually, few studies have integrated these dimensions within a coherent, multi-level analytical framework. This lack of integration has resulted in fragmented explanations that fail to capture the complex interplay of structural conditions, environmental pressures, and enabling institutions. Consequently, policy responses derived from such partial understandings may be less effective in addressing the real constraints faced by farmers on the ground. Fourth, behavioral and psychological determinants—such as risk perception, time preferences, trust in institutions, and intra-household dynamics—are frequently overlooked or only superficially addressed in adaptation studies. Yet these factors critically shape how individuals interpret climate risks, evaluate adaptation options, and commit to action. In particular, farmers' subjective expectations about future climate trends, their past experiences with crop failure, or the influence of gendered power relations within households can significantly influence whether and how adaptation occurs. The absence of these elements from most quantitative analyses leaves an important explanatory gap in understanding why some households adapt while others do not, even when faced with similar climatic and economic conditions [44,45].

From the analysis of the context and gaps in previous studies, this study proposes three main research questions (RQ).

**RQ1.** *What are the key socio-economic, institutional, and environmental factors that shape the heterogeneous adoption of climate change adaptation strategies among farming households in coastal and salinity-prone areas of transitional economies like Vietnam?*

**RQ2.** *How do behavioral attributes—such as risk perception, farming experience, and intra-household dynamics—interact with structural constraints to influence the selection of specific adaptation strategies to saltwater intrusion and drought?*

**RQ3.** *Based on empirical evidence from multivariate modeling, what context-specific and scalable policy recommendations can be formulated to enhance the adaptive capacity of smallholder farmers facing climate-induced risks in Vietnam's coastal regions?*

This study makes some novelties and contributions to the literature on climate change adaptation, particularly within the context of transitional economies in Southeast Asia. First, it provides one of the few empirical investigations into household-level adaptation strategies in coastal and salinity-prone regions of Vietnam—an area highly vulnerable to climate-induced hazards yet underrepresented in current academic discourse. Second, the study advances methodological rigor by employing a Multinomial Logit (MNL) model to capture the differentiated nature of adaptation decisions across multiple strategies, moving beyond aggregate or binary analyses commonly used in prior research. Third, it integrates a multidimensional framework that simultaneously considers socio-economic, demographic, institutional, and environmental determinants, offering a more holistic understanding of adaptation behavior. Fourth, the study addresses the often-overlooked behavioral and psychological aspects of adaptation, including gender dynamics, experience, and perception of risk, which are critical in shaping real-world decision-making under uncertainty. Fifth, by disaggregating adaptation responses to specific stressors such as saltwater intrusion and drought, the research provides context-sensitive insights that can inform localized adaptation planning. Finally, the study contributes practical value by generating evidence-based policy recommendations tailored to different household profiles, thereby supporting the development of inclusive and scalable adaptation strategies aligned with national climate resilience goals.

## 2. Materials and methods

### 2.1 Study site

This study was conducted in Quang Binh province in the Central region of Vietnam, specifically in Tuyen Hoa district (17.7882° N, 106.2051° E) in the upper reaches of the Gianh River, which has been severely affected by drought and saltwater intrusion over the past 10 years. Gianh River is originated from the 2,017 m high Co Pi mountainside area in the Truong Son range, flowing through the districts of Minh Hoa, Tuyen Hoa, Bo Trach, Quang Trach, and Ba Don commune, then going to the East Sea at Cua Gianh. The Gianh River upstream is located in the northwest of Tuyen Hoa district, from the mountains on the Lao Bao peninsula. Mountainous terrain and steep hills, with small streams, waterfalls, and lakes, characterize this area. In the Tuyen Hoa district, the Gianh River upstream flows through the production areas of three communes Chau Hoa, Phong Hoa, and Duc Hoa [46,47]

Tuyen Hoa is always affected by northern air masses in winter (Southeast wind) and hot, dry West winds in summer (Laos wind). The average annual temperature ranges from 24°C - 25°C, the average annual rainfall is about 2,100–2,300 mm and is divided into two distinct seasons: the rainy season and the low-rainy season. The rainy season is from September to March of the following year, the average temperature is from 22°C - 23°C, the lowest temperature is in January (about 10°C – 14°C) [48]. Rainfall is concentrated in September, October and November, accounting for 70% − 80% of the total annual rainfall reaching 2,000–2,300 mm, due to the influence of the northeast monsoon combined with disturbances that cause heavy rain such as storms and high pressure. Tropical lows, tropical convergence zones, leading to flooding in the delta. The rainy season is from April to August, the average temperature is about 26°C −27°C. The hottest month can reach 39°C (June, July and August), the average rainfall only accounts for 20% − 30% of the total annual rainfall [35,47].

The economy of Tuyen Hoa district and the three above communes is mainly based on agriculture. In particular, water from the river has played an essential role in developing agriculture and enriching the locality. This is also where

people rely on the Gianh River for daily life and agricultural production. The river water is also used for irrigation, aquaculture, and providing domestic water for the community. According to Mai and Truong (2022), with current changes in weather and environment, the Gianh River upstream area is facing challenges regarding water resources. The prolonged heatwave and increasing seawater intrusion have also caused unwanted impacts on the lives and production of the communities. Reality shows that heatway and saltwater intrusion incidents have increased significantly in recent years compared to the period 2000–2010 and usually occur from April to July every year. These two types of disasters often come together and affect agricultural livelihoods. Firstly, it affects aquaculture: increases the cost of aquaculture activities (if salinity increases, farmers must add sugar to the water or pump more fresh water to neutralize and reduce pond salinity to < 25%). If the phenomenon of saltwater intrusion occurs for a long time, farmers cannot pump water into the pond, causing the shrimp to get sick (because there is not enough oxygen to breathe), so aquaculture output can decrease and cause total loss. Second, it affects farming activities: making the soil saline and acidic, rice plants are unable to grow, productivity decreases by 60–70%, people cannot cultivate and convert to shrimp farming ponds, brackish water crab or rice fish have higher economic efficiency, most of the land area is in low-lying areas. In addition, drought causes lack of water to irrigate the rice growing area, mainly in the Winter-Spring crop. Planted rice having not enough irrigation water will might reduce productivity. If the drought lasts about 1 month, it can cause crop failure (due to dead rice or flat rice grains) [35,46–48].

### 2.2 Empirical model development

Adaptation has been widely acknowledged as a critical pillar in the policy response to climate change, particularly in the context of agricultural resilience and rural livelihoods [23,49]. A substantial body of literature has examined the ways in which various regions—especially in developing countries—respond to climate-induced challenges, as well as the key factors shaping these adaptive behaviors [50–52]. Existing studies have demonstrated that farmers' adaptation strategies are highly context-specific and influenced by a range of factors, including but not limited to farm size, land type, agroclimatic conditions, and broader ecological, institutional, socio-economic, cultural, and political contexts [41,42,50,52]. More recent analyses have identified a complex interplay of determinants that extend beyond structural factors to include institutional accessibility, political frameworks, social networks, cultural norms, behavioral and cognitive dimensions, and gender dynamics [53]. Despite the growing interest in climate change adaptation within the Global South, including parts of Asia and Africa, there remains a notable research gap concerning the specific barriers and enabling conditions for adopting adaptive strategies in Vietnam, particularly at the household level [44,45,54]. In response to this gap, the present study investigates the factors influencing farming households' decisions to adopt climate variability adaptation strategies in Vietnam. The analysis encompasses a range of determinants across three broad categories: (i) socio-economic variables such as farm size, income levels, civic organization membership, and farming experience; (ii) demographic characteristics including household size and age of the household head; and (iii) institutional factors such as access to credit, markets, technical training, and climate-related information. These factors are conceptualized in the analytical framework summarized in Fig 1, which illustrates the multidimensional nature of adaptation decision-making among rural households.

To identify essential variables in farmers' choice of adaptation strategies, we performed a multinomial logit analysis model (MNL) as previous studies [55–58]. The model allowed determining the relationship between the probability of choosing an adaptation option and a set of explanatory variables, then coming up with the determination of probabilities of different selected adaptive options [59].

In MNL model, let Y be a random dependent variable with values {1 to J} represents the adaptation strategies chosen by farming households (1 to J) and let $X_i$ be the set of variables described as socio-economic characteristics, institutional factors, and climatic properties. To show how changes in the elements are determined by summing the probabilities of $P(Y_i) = \frac{j}{X_i}$. The probability of a dependent variable belonging to the $n^{th}$ category in a multinomial logistic regression model is expressed as below [59,60].

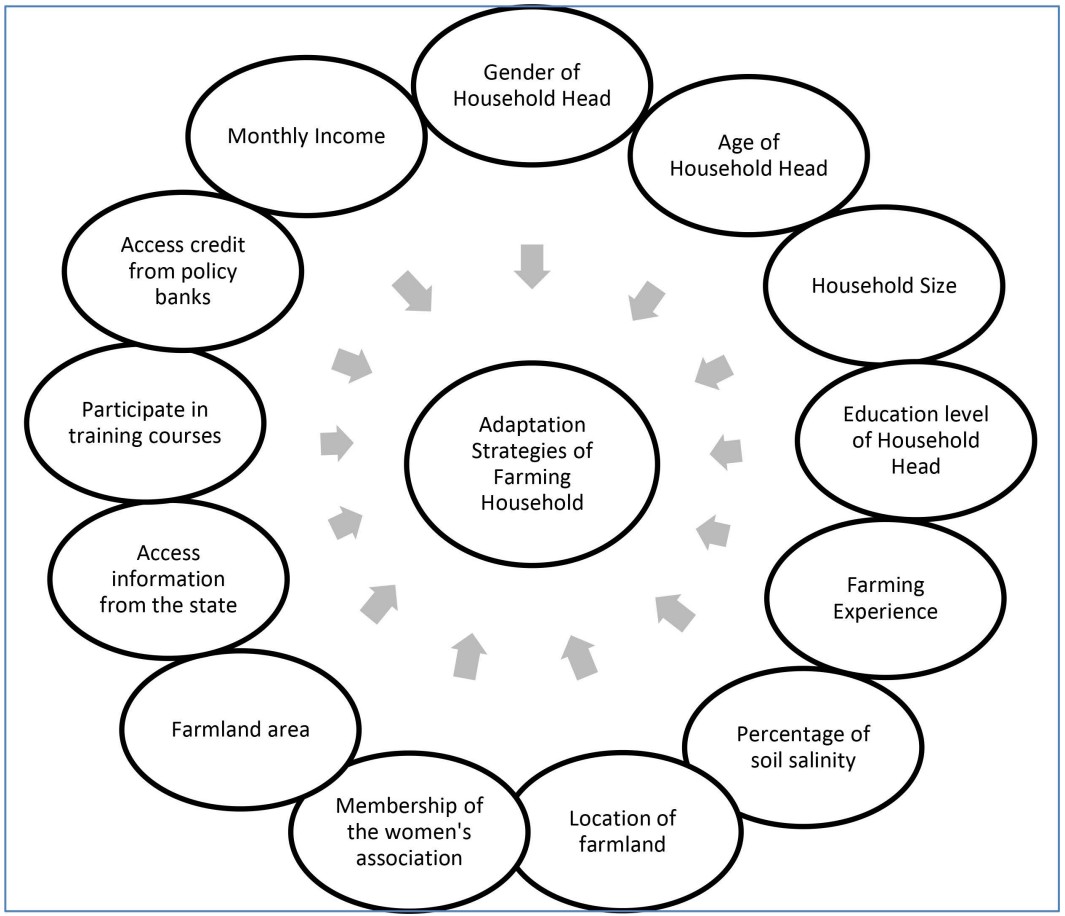

**Fig 1. Proposed study model of impact factors.** Source: Study design (2024).

$$\pi j = \frac{exp(\sum_{k=1}^{K} \beta jkxk)}{1 + \sum_{j=1}^{J-1} \beta jkxk} \; with \; j = 1, 2, \ldots . J - 1$$

(1)

In models with dependent variables of more than two types, it is necessary to identify a base type for comparison or analysis. The base category (J) can be chosen as 0 for the dependent variable and consisting of categories 0, 1, 2, 3, and 4. The model was linearized by taking the natural logarithm of these odds ratios to obtain logistic models. If J is chosen as the base type, then the probability of the dependent variable being in the base type is determined as given in the equation below.

$$\pi j = P(Y = J) = \frac{1}{1 + \sum_{j=1}^{J-1} exp[(\sum_{K=1}^{K} \beta jkxk)]} \; with \; j = 1, 2, \ldots . J - 1$$

(2)

The overall relationship between an individual independent variable and the dependent variables were tested by testing the coefficient – 2likelihood based on summing the probabilities associated with the predicted and actual outcomes.

$$-2LogLikelihood = \sum_{n=i}^{N} ([Y_i lnP(Y_i)) + (1 - Y_i)$$

(3)

The marginal effect of exogenous factors is given by:

$$\delta_j = \frac{P_j}{X_k} = P_j\left[\beta_{jk} - \sum_{J=1}^{J-1} P_j\beta_{jk}\right] = P_j(\beta_j - \beta)$$

(4)

The marginal effect measures the expected change in the probability of a particular choice being made for a unit change in the explanatory variable. The signs of the marginal effects and the corresponding coefficients could be different because the coefficients depended on the signs and magnitudes of all the other coefficients [61–64]. Table 1 described the variables expected to be included in the model to analyse factors affecting the choice of livelihood strategies to adapt to climate change of farming households. The adoption of these strategies was hypothesized to be influenced by demographic and socio-economic characteristics, such as gender, age of the household head, household size, education level, farm size, annual income, and number of livestock and breeds owned; institutional factors, such as credit services, agricultural extension services, social networks, access to information and awareness of climate change.

## 2.3 Data collection and analysis

This study employed both primary and secondary data sources to ensure comprehensive and contextually grounded analysis of the factors influencing climate change adaptation strategies among farming households. Primary data were collected through a combination of qualitative and quantitative methods. Specifically, focus group discussions and in-depth interviews were conducted with local officials, commune leaders, and villagers to explore community-level perceptions and challenges associated with climate change adaptation. These methods were instrumental in contextualizing the research, refining the questionnaire design, and capturing localized knowledge and lived experiences that quantitative data alone could not fully reveal. The qualitative insights also informed the selection of variables for subsequent modeling and analysis. In addition to field-generated data, secondary data were collected from several authoritative sources to complement

**Table 1. Description of explanatory variables in the model.**

| No. | Explanatory variables | Code | Expected signs | Measurement |
|---|---|---|---|---|
| 1 | Gender of respondent | GEN | +/- | 1: male 0: female |
| 2 | Household size | SIZE | +/- | Continuous |
| 3 | Education level of respondent | EDU | +/- | Continuous |
| 4 | Age of respondent | AGE | +/- | Continuous |
| 5 | Household monthly income | INC | +/- | Continuous |
| 6 | Farming experience | EXP | +/- | Continuous |
| 7 | Percentage of soil salinity | SAL | +/- | Continuous |
| 8 | Location of farmland | POS | +/- | Continuous |
| 9 | Farmland area | ARE | +/- | Continuous |
| 10 | Access information from the state | INF | + | 1: yes; 0: no |
| 11 | Membership of the local associations | MEM | + | 1: yes; 0: no |
| 12 | Access credit from policy banks | CRE | + | 1: yes; 0: no |
| 13 | Participate in training courses | TRA | + | 1: yes; 0: no |

Source: Study design (2024).

and validate the primary findings. These included socio-economic and climatic data Quang Binh Statistics Department, and the Department of Climate Change under the Ministry of Natural Resources and Environment (MONRE). These sources provided critical background information on agricultural practices, historical climate variability, and household demographics in the study area.

The sample size for the household survey was determined using a simplified formula recommended by Greene (2000) [65] expressed as:

$$n = \frac{N}{1 + N * e^2}$$

(5)

where n represents the required sample size, N is the total population, and e is the allowable margin of error, set at 5% in this study. Based on official statistics from the Quang Binh Statistics Office (2021) [48], the combined population of the three selected communes—Chau Hoa, Phong Hoa, and Duc Hoa— were approximately 14,521 individuals. With an average household size of 4.5 persons per household in rural areas [46], the estimated total number of households was 3,226. Applying the above formula yielded a required minimum sample size of 356 households to ensure statistical reliability and representativeness. The three communes were purposefully selected due to their high concentration of farming households and significant exposure to climate risks [46,48]. Given the relatively even distribution of households across the selected sites, a proportional random sampling approach was adopted. Accordingly, 119 households were surveyed in Phong Hoa, 119 in Duc Hoa, and 118 in Chau Hoa. Lists of households were obtained from communal authorities, from which respondents were selected using simple random sampling techniques to minimize selection bias.

To ensure ethical compliance and participant engagement, surveys were conducted in the evenings when household members were more likely to be present. Prior to the interviews, each participant was provided with a clear explanation of the study's objectives and procedures. Informed consent was obtained in writing through a consent form, in which participants voluntarily indicated their willingness to participate and signed their names. All selected households agreed to take part in the study. The official survey was conducted over a two-month period, from November to December 2023.

Upon completion of data collection, the survey responses were screened for completeness and consistency. Cleaned data were then coded and entered into the Statistical Package for the Social Sciences (SPSS) version 23.0 for quantitative analysis. The initial phase of analysis included descriptive statistics, frequency distributions, and cross-tabulations to summarize household characteristics and assess general attitudes and perceptions regarding climate change and its effects on agricultural livelihoods. ANOVA tests were also conducted to examine differences in awareness and adaptive behavior across socio-economic subgroups. To identify the key determinants of adaptation strategy choices, the study employed the MNL model. This model was appropriate given the categorical nature of the dependent variable—households' choice of specific adaptation strategies—and allowed for the estimation of the likelihood that a household selects a particular strategy given a set of independent socio-economic, demographic, and institutional variables. The MNL results provided empirical evidence on the significance and direction of each factor's influence on adaptive decision-making in the context of climate-related stressors.

## 3. Results

### 3.1 Socio-economic characteristics of the sample

Table 2 presents the socio-economic characteristics of the sampled farming households. The gender distribution of household heads indicated that 59.78% were male and 40.22% were female, suggesting that male-headed households constituted the majority of the sample. The average household size was 5.23 members. Larger households were observed to utilize a broader range of livelihood resources, derive greater income from environmental activities, and exhibit a higher likelihood of adopting climate change adaptation strategies compared to smaller households. The age of the household

**Table 2. Socio-economic description of the sample.**

| Variables | Categories | Frequency (N = 356) | Percentage | Means |
|---|---|---|---|---|
| Gender | Female | 172 | 40.22 | – |
| | Male | 184 | 59.78 | |
| Age | Young | 99 | 27.81 | 45.24 |
| | Adulthood | 231 | 64.89 | |
| | Old | 26 | 7.30 | |
| Household size (person) | 1–4 | 153 | 42.98 | 5.23 |
| | 5–8 | 149 | 41.85 | |
| | 9–12 | 54 | 15.17 | |
| Education level | Illiterate | 53 | 14.89 | – |
| | Primary | 109 | 30.62 | |
| | Secondary | 116 | 32.58 | |
| | High school | 48 | 13.48 | |
| | College and above | 30 | 8.43 | |
| Landholding (hectare) | Less than 3 | 251 | 70.51 | 1.65 |
| | More than 3 | 105 | 29.49 | |
| Livestock holding (kg) | 0-1,000 | 331 | 92.98 | 1,363 |
| | More than 1,000 | 25 | 7.02 | |
| Monthly income | Below 10 million VND | 179 | 50.28 | 16,890,560 |
| | 11−30 million VND | 156 | 43.82 | |
| | More than 30 million VND | 21 | 5.90 | |

Source: Study results (2024).

head was employed as a proxy for farming experience. It was found that younger household heads generally possessed less familiarity with the socio-ecological context and agricultural practices than their older counterparts. Age categories were established based on standard psychomotor developmental criteria, classifying respondents into three groups: young, adult, and elderly. The results showed that the majority of respondents fell into the adult category, with an average age of 45.24 years. This demographic profile is advantageous for the study, as middle-aged individuals are likely to have accumulated substantial experiential knowledge about local climate variability, its impacts, and relevant adaptation practices.

Education was found to play a significant role in enhancing adaptive capacity. Educated respondents demonstrated a higher propensity to adopt adaptation measures, likely due to improved access to information, technology, and diversified income sources. Survey results revealed that 14.89% of respondents were illiterate, while 85.11% were literate. This illiteracy rate is relatively high compared to national averages, consistent with patterns observed in other rural areas of Quang Binh province. The high proportion of illiterate farmers may hinder their capacity to respond effectively to climate change due to limitations in accessing adaptation-related knowledge and technologies. Farm size, defined as the total area of agricultural land owned by a household, was measured in hectares. The data indicated that land ownership ranged from 0.1 to more than 3 hectares, with an average of 1.65 hectares per household. Livestock ownership was used as a proxy for household wealth, given that livestock remains a critical indicator of economic status in rural areas. The average livestock holding across sampled households was equivalent to 1,363 kilograms in terms of stockholding weight. Monthly household income was also assessed as a key economic indicator. The mean monthly income of surveyed households was 16,890,560 VND (approximately 700 USD). These figures reflect the relatively low-income status of the sampled population and underscore the economic vulnerability of farming households to climate-related shocks. Institutional factors

were also examined, particularly in terms of access to agricultural extension services and weather-related information. The findings indicated that 78.21% of households had access to agricultural extension services, while 81.34% reported receiving information on weather and climate conditions. These institutional supports are considered crucial for enhancing farmers' capacity to anticipate and respond to climatic variability and extreme events.

### 3.2 Livelihood vulnerability to climate change and adaptation strategies by households

The present study investigated the level of awareness and the types of adaptation strategies adopted by farming households in response to disasters induced by climate change in the local context of Quang Binh province. A key focus was placed on understanding how communities perceive climatic changes and how this perception informs their coping and adaptive behaviors. The findings reveal that a substantial majority—approximately 85.23% of surveyed households— demonstrated an awareness of the ongoing manifestations of climate change and the increasing frequency of extreme weather events occurring in their locality. This high rate of awareness suggests that climate change and its impacts have become observable and tangible realities for the local population.

Households reported that the most commonly experienced climate-related hazards in the past five years include floods, tropical storms, saltwater intrusion, and drought. These hazards have not only occurred with increasing frequency but also with greater intensity and duration, posing heightened threats to agricultural livelihoods. When asked to compare the frequency of such disasters to previous years (with 2018 serving as a reference point), 51.9% of respondents perceived that droughts had become more frequent, followed by 42.2% who observed an increase in flooding, and 31.3% who reported a rise in saltwater intrusion. Interestingly, although only 36.6% of respondents noted an increase in storm frequency, 63.4% believed that the frequency of storms had decreased. Nonetheless, the majority emphasized that the *intensity* of individual storms has significantly escalated. For example, the storm Wutip, which struck in October 2018, was cited by multiple respondents as particularly destructive, underscoring the need to assess not only the occurrence but also the severity of climatic hazards (Table 3).

Respondents further identified the temporal patterns of specific climate-induced disasters. Flooding is typically observed from August to October and may extend into November, often associated with the typhoon season. In contrast, droughts and saltwater intrusion are most prevalent between April and July, with the peak intensity occurring between April and June. During this period, the lack of rainfall combined with high temperatures and salinity intrusion severely undermines rice production—the primary livelihood activity in the area. Interviewees consistently reported that saltwater intrusion, when combined with drought, causes substantial losses in rice yields, particularly in low-lying paddy fields with poor irrigation infrastructure. These conditions significantly disrupt planting calendars, reduce agricultural productivity, and increase the risk of long-term livelihood insecurity. The seasonal concurrence of storm-induced flooding and drought-related phenomena places significant pressure on the already limited adaptive capacity of local communities. In the months of August through November, storms often result in flash floods, damaged irrigation systems, and waterlogging of crops. Conversely, during the dry season from April to July, extended dry spells and salinity intrusion compromise water

**Table 3. Perception of types of accidents occur every month of the year.**

|  | Unit: % | | | | | | | | | | | |
| --- | --- | --- | --- | --- | --- | --- | --- | --- | --- | --- | --- | --- |
| Month | 1 | 2 | 3 | 4 | 5 | 6 | 7 | 8 | 9 | 10 | 11 | 12 |
| Salinization | 2.1 | 3.2 | 2.3 | 15.8 | 20.2 | 25.5 | 10.3 | 5.1 | 8.1 | 7.4 | 8.3 | 2.1 |
| Drought | 0 | 0 | 0.8 | 26.2 | 66.1 | 88.3 | 54.4 | 6.3 | 0 | 0 | 0 | 0 |
| Flooding | 0 | 0 | 0 | 0 | 0 | 0 | 2.12 | 43.1 | 93.2 | 72.6 | 14.1 | 0.8 |
| Storm | 0 | 0 | 0 | 0 | 0 | 0 | 0 | 20.5 | 86.4 | 80.7 | 56.5 | 1.2 |

Source: Study results (2024).

availability for irrigation and domestic use, further compounding vulnerability. The spatial and temporal overlap of multiple stressors necessitates context-specific adaptation strategies tailored to distinct hazard profiles across different times of the year.

Importantly, the subjective observations of the surveyed population regarding changing climate patterns and disaster risks align closely with national assessments and projections. The observed trends in increased drought severity, prolonged flood seasons, and rising storm intensity are consistent with the findings outlined in Scenario of climate change, sea level rise for Vietnam [27]. This convergence between community-level perceptions and scientific projections lends further credibility to the argument that climate change has transitioned from a theoretical concern to a lived experience among rural households in Vietnam.

The insights gained from this study emphasize the importance of localized climate knowledge and the role of community-based observations in informing adaptation planning. They also highlight the necessity of enhancing farmers' adaptive capacity through timely dissemination of climate information, disaster preparedness education, and investment in resilient agricultural infrastructure. The findings provide a foundation for future interventions aimed at improving resilience and fostering sustainable rural development in the face of accelerating climatic threats.

To respond effectively to the multifaceted impacts of climate change, a notable proportion of surveyed farming households—approximately 42.13%—reported having adopted at least one or more adaptation strategies tailored to their specific livelihood conditions. The descriptive statistical analysis revealed a diversified array of coping mechanisms utilized by farmers in Tuyen Hoa district, reflecting both agricultural innovation and livelihood diversification. Among the identified adaptation practices, shrimp farming emerged as the most prevalent strategy, adopted by 76.69% of households. This was followed by lotus-fish integrated farming systems (64.33%), and the cultivation of vegetables (55.9%) as alternative or supplementary sources of income and food security. Furthermore, 66.85% of respondents reported switching to new salt-tolerant rice varieties—a crucial adaptation in the context of increasing soil salinity driven by saltwater intrusion.

In terms of technological and infrastructural adaptation, approximately 23.88% of households had installed small-scale irrigation systems, which, although limited in coverage, are essential in areas with erratic rainfall and unreliable surface water availability. Adjustments in agricultural calendars also constituted a common adaptation approach, adopted by 49.72% of households. These adjustments were made to synchronize agricultural activities with seasonal weather variations. Specifically, aquaculture activities were typically initiated between February and May (lunar calendar), the winter-spring rice crop was planted from late December to April, and the summer-autumn crop was sown from mid-May to mid-August annually. Additionally, 34.83% of respondents indicated participation in non-agricultural employment as a strategy to diversify income sources and reduce dependence on climate-sensitive sectors (Table 4).

Table 4.  Adaptation strategies of farming households against saltwater intrusion.

| Adaptation strategies | n | % of respondents |
|---|---|---|
| Shrimp farming | 273 | 76.69 |
| Lotus and fish farming | 229 | 64.33 |
| Vegetable production | 199 | 55.90 |
| Small-scale irrigation system | 85 | 23.88 |
| Salt-tolerant rice varieties | 238 | 66.85 |
| Agricultural calendar changing | 177 | 49.72 |
| Non-agricultural work participation | 124 | 34.83 |

Source: Study results (2024).

### 3.3 Challenges for climate change adaptation

To better understand the constraints to adaptation implementation, farmers were asked to identify and rank key challenges that hinder the effective adoption of these strategies. This evaluation was conducted using a 5-point Likert scale ranging from 1 ("completely disagree") to 5 ("strongly agree"). The mean scores of each statement were calculated and ranked accordingly to reflect the perceived severity of the challenges.

The results of the ranking exercise highlighted "lack of water for irrigation" as the most critical constraint. This issue was consistently emphasized by respondents as a major barrier to sustaining agricultural production, particularly in the context of coastal Vietnam, where freshwater scarcity is exacerbated by climate-induced phenomena such as prolonged droughts, saltwater intrusion, and river salinization. Given the reliance of farming systems on water sources—rivers, rainfall, canals, and groundwater—any disruption to water availability significantly undermines productivity and food security. Although governmental organizations (GOs), non-governmental organizations (NGOs), and local civic groups have engaged in efforts to address water management issues, the challenges remain pervasive and unresolved in many areas.

The second highest-ranked constraint was "lack of cultivable land." With ongoing demographic expansion and rural-urban transformation, increasing amounts of farmland are being converted to non-agricultural uses, such as housing development, industrial infrastructure, and public facilities. This trend of agricultural land loss poses a serious threat to climate resilience, as it reduces the physical space available for adaptation-oriented diversification and sustainable farming practices.

The third most frequently cited constraint was "lack of capital." Financial limitations were reported by 48% of the respondents as a significant obstacle preventing the adoption of adaptation strategies. Limited access to credit and low savings rates reduce farmers' ability to invest in climate-resilient technologies, improved seeds, irrigation systems, and other necessary inputs. In this context, access to microfinance, rural credit services, and public subsidies may play a critical role in facilitating wider adaptation adoption.

Other identified barriers included "lack of information" and "lack of technology," which were noted by approximately 32% and 30% of the respondents, respectively. These constraints highlight the role of institutional support and information dissemination in climate adaptation processes. Without timely access to reliable climate forecasts, agricultural extension services, or demonstrations of new technologies, farmers are likely to rely on traditional practices that may no longer be sufficient under changing environmental conditions.

In summary, the findings emphasize that adaptation to climate change among smallholder farmers in Central Vietnam is constrained by a combination of physical, financial, and informational barriers. Addressing these barriers—particularly those related to irrigation water, land availability, capital resources, access to climate information, and agricultural technologies—is essential for enhancing the adaptive capacity of rural communities. Doing so will contribute not only to the resilience of farming systems but also to the broader goals of sustainable agricultural development and rural poverty reduction in Vietnam (Fig 2).

### 3.4 Factors influencing households' choice of climate change adaptation strategies

To investigate the determinants influencing the adoption of climate change adaptation strategies among farming households in the study area, a MNL model was employed. This model is particularly suitable for analyzing discrete choice data where the dependent variable represents the selection among multiple, unordered alternatives. In this study, the dependent variable was defined as the household's choice of one of four main adaptation strategies: (1) vegetable production, (2) shrimp farming, (3) adoption of salt-tolerant rice varieties, and (4) lotus-fish integrated farming. Each of these strategies reflects a distinct approach to mitigating the adverse effects of climate variability and environmental stressors specific to the local agro-ecological context.

For robustness of analysis and binary classification purposes, these four adaptation strategies were pooled, and an index-based transformation was applied. Specifically, households that adopted adaptation strategies at a level above the

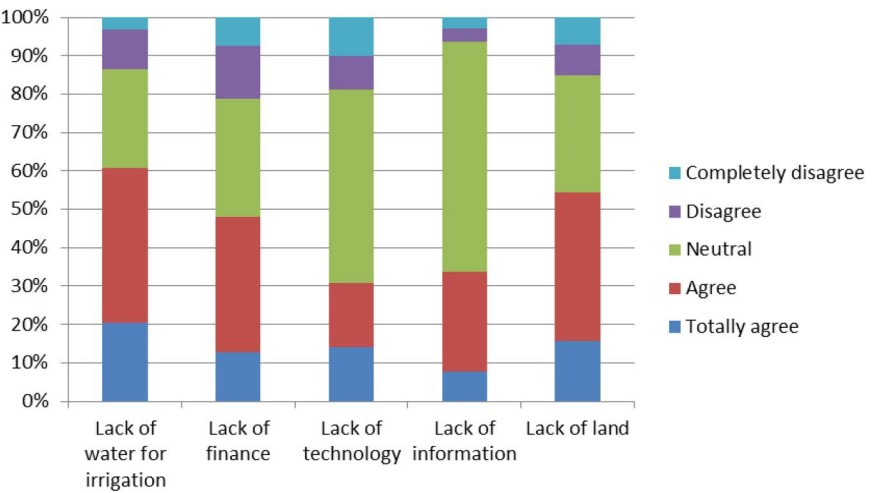

**Fig 2. Challenges for climate change adaptation of farmers.** Source: Study results (2024).

sample mean were coded as '1' (i.e., indicating active application of adaptive strategies), while those adopting strategies at or below the average level were coded as '0' (i.e., no or limited adaptation). This dichotomization facilitated the identification of significant factors distinguishing adaptive versus non-adaptive households.

Prior to estimating the MNL model, the potential multicollinearity among independent variables was assessed using Variance Inflation Factor (VIF) [65]. All VIF values were found to be below the threshold value of 2.0, indicating an absence of serious multicollinearity concerns. This confirms that the explanatory variables included in the model were sufficiently independent of one another, ensuring the reliability of coefficient estimates. Model adequacy and overall goodness-of-fit were assessed through the Likelihood Ratio (LR) Chi-square test and Pearson Chi-square statistics. The LR Chi-square value was 81.032, statistically significant at the 1% level (p < 0.01), suggesting that the explanatory variables jointly contribute to the variation in adaptation strategy choices and that the model significantly improves upon the null model with no predictors. Furthermore, the Pearson Chi-square statistic for goodness-of-fit was 182.342, also highly significant (p < 0.01), providing strong evidence that the model has a good fit to the observed data and demonstrates adequate explanatory power.

The empirical results from the MNL model indicate that the selection of climate change adaptation strategies among farming households is not random but is systematically influenced by a range of factors. These include household-level socio-economic characteristics (such as education, income, household size, and farming experience), institutional factors (such as access to extension services, information on climate change, and participation in civic organizations), organizational capacity (such as access to credit and irrigation infrastructure), and environmental characteristics (including land-holding size, frequency of extreme weather events, and water availability). Each of these variables exerts a differentiated impact on the probability of selecting a specific adaptive strategy, reflecting the heterogeneity in household capacities and climate exposures. These findings underscore the importance of designing context-specific adaptation policies that are sensitive to the underlying socio-economic and environmental drivers influencing household behavior. In particular, strengthening institutional support systems, enhancing access to agricultural services and information, and addressing resource-related constraints (e.g., water, land, and capital) are crucial for promoting the adoption of effective and sustainable adaptation practices at the grassroots level (Table 5).

Specifically, the findings in Table 6 indicated that gender (GEN) significantly influenced adaptation choices. Male-headed households were more likely to adopt shrimp farming (+31.6%), lotus-fish farming (+41.2%), and salt-tolerant rice

**Table 5. Results of MNL analysis.**

| Variables | Vegetable production | | | Shrimp farming | | | Salt-tolerant rice varieties | | | Lotus and fish farming | | | VIF |
|---|---|---|---|---|---|---|---|---|---|---|---|---|---|
| | B | Sig. | Exp(B) | B | Sig. | Exp(B) | B | Sig. | Exp(B) | B | Sig. | Exp(B) | |
| Intercept | 0.234 | 0.912 | | 0.059 | 0.932 | | 9.783 | 0.00 | | 4.534 | 0.032 | | |
| GEN | −0.354 | 0.567 | 0.745 | 1.321 | 0.042** | 2.987 | 1.732 | 0.003*** | 6.324 | 0.989 | 0.312** | 3.313 | 1.83 |
| SIZE | −0.301 | 0.432 | 0.789 | −0.112 | 0.789 | 0.981 | 1.564 | 0.000*** | 4.324 | 0.645 | 0.045** | 3.432 | 1.64 |
| EDU | 1.623 | 0.000*** | 4.021 | −0.708 | 0.071** | 0.645 | 0.608 | 0.067** | 1.785 | −0.211 | 0.645 | 0.678 | 1.46 |
| AGE | −0.178 | 0.345 | 0.843 | −0.213 | 0.654 | 0.708 | 0.432 | 0.078** | 0.822 | −0.032 | 0.756 | 0.867 | 1.75 |
| INC | 0.702 | 0.065** | 2.019 | 0.205 | 0.322 | 1.453 | 0.004 | 0.998 | 1.323 | 0.322 | 0.876 | 1.423 | 1.58 |
| EXP | 0.342 | 0.424 | 1.432 | −0.599 | 0.323 | 0.607 | 0.199 | 0.798 | 1.543 | −0.543 | 0.321 | 0.876 | 1.44 |
| SAL | −0.601 | 0.587 | 0.643 | −0.132 | 0.876 | 0.895 | 2.567 | 0.061* | 10.342 | 1.432 | 0.312 | 2.764 | 1.39 |
| POS | −0.089 | 0.832 | 0.912 | 0.673 | 0.069** | 2.868 | −0.345 | 0.436 | 0.654 | 0.312 | 0.543 | 1.312 | 1.28 |
| ARE | −0.243 | 0.656 | 0.802 | 0.883 | 0.102** | 2.124 | 1.812 | 0.000*** | 5.756 | 1.312 | 0.032** | 2.977 | 1.74 |
| INF | −0.061 | 0.912 | 0.899 | 0.064 | 0.899 | 1.069 | 0.032 | 0.998 | 1.434 | −0.433 | 0.534 | 0.677 | 1.49 |
| MEM | −0.312 | 0.632 | 0.657 | −0.073 | 0.876 | 0.911 | 0.045 | 0.997 | 1.543 | −0.534 | 0.123 | 0.656 | 1.63 |
| CRE | −0.189 | 0.872 | 0.799 | 1.321 | 0.065** | 3.567 | 0.732 | 0.245 | 1.432 | 0.766 | 0.032** | 2.543 | 1.81 |
| TRA | 0.005 | 0.902 | 1.021 | 0.991 | 0.117 | 2.563 | 0.321 | 0.534 | 1.765 | 0.432 | 0.334 | 1.876 | 1.78 |
| Number of observations | 356 | | | | | | | | | | | | |
| LR chi-square | 81.032* | | | | | | | | | | | | |
| Log Likelihood | −398.214 | | | | | | | | | | | | |
| Pseudo-R square (McFadden) | 0.6193 | | | | | | | | | | | | |
| Goodness-of-Fit (Chi-square)/Pearson/ | 182.342 | | | | | | | | | | | | |

***, **: significant at 1% and 5% error levels, respectively.

Source: Study results (2024).

**Table 6. Marginal effects from MNL model of adaptation strategies.**

| Explanatory variables | Code | Vegetable production | Shrimp farming | Salt-tolerant rice varieties | Lotus-fish farming |
|---|---|---|---|---|---|
| Gender | GEN | 1.324 | 0.316** | 0.087*** | 0.412** |
| Household Size | SIZE | 0.005 | 0.042 | −0.097*** | 0.071* |
| Education level | EDU | 0.021 | −0.058*** | −0.177*** | 0.096** |
| Age | AGE | 0.007 | −0.007** | 0.005** | −0.002*** |
| Monthly income | INC | −0.032** | 0.039*** | 0.067*** | 0.001*** |
| Farming experience | EXP | 0.011** | −0.015*** | 0.061*** | 0.004*** |
| Percentage of soil salinity | SAL | 0.031*** | 0.086*** | −0.008** | 0.056*** |
| Location of farmland | POS | −0.061*** | −0.093* | 0.054*** | 0.011 |
| Farmland area | ARE | 0.026 | 0.002*** | −0.043** | 0.042** |
| Access information from the state | INF | 0.041 | 0.105* | 0.121** | 0.031 |
| Membership of the local associations | MEM | 0.098** | 0.123** | 0.342** | 0.046* |
| Access credit from policy bank | CRE | 0.014 | 0.008** | 0.005 | 0.006** |
| Participate in training courses | TRA | 0.042*** | 0.086*** | 0.341 | 0.037*** |

***, **: significant at 1% and 5% error levels, respectively.

Source: Study results (2025).

varieties (+8.7%) than female-headed households. Similarly, household size (SIZE) showed differentiated effects: while increasing household size decreased the probability of adopting salt-tolerant rice varieties by 9.7%, it increased the probability of lotus-fish adoption by 7.1%.

Education level (EDU) exhibited mixed effects. Higher educational attainment negatively impacted the likelihood of adopting shrimp farming (−5.8%) and salt-tolerant rice (−17.7%) but positively influenced the adoption of lotus-fish farming (+9.6%), likely due to the technical complexity of this method.

Age (AGE) was positively correlated with the adoption of salt-tolerant rice varieties (+0.5% per year) but negatively associated with shrimp (−0.7%) and lotus-fish farming (−0.2%), which require higher physical labor and technical adaptation. No quadratic relationship with age squared was observed.

Monthly income (INC) positively influenced the adoption of shrimp farming (+3.9%), salt-tolerant rice (+6.7%), and lotus-fish (+1%) models, while it had a negative effect on vegetable production (−3.2%).

Farming experience (EXP) had a positive impact on vegetable production (+1.1%), lotus-fish (+0.4%), and salt-tolerant rice (+6.1%) but reduced the likelihood of shrimp adoption (−1.5%).

The proportion of salinized land (SAL) was positively associated with shrimp farming (+0.6%), vegetable (+3.1%), and lotus-fish farming (+5.6%), while it reduced the adoption probability of new rice varieties (−0.8%). Similarly, the current position of land affected by salinity (POS) increased rice adoption (+5.4%) but decreased shrimp (−9.3%) and vegetable (−6.1%) uptake.

Agricultural land area (ARE) positively influenced the adoption of shrimp (+0.2%) and lotus-fish (+4.2%) methods, but negatively affected rice variety adoption (−4.3%), suggesting that land-extensive methods like shrimp farming are more feasible for larger landholders.

Access to climate information (INF) increased adoption probabilities of salt-tolerant rice (+12.1%) and shrimp farming (+10.5%). Membership in civic organizations (MEM) significantly enhanced the probability of adopting all strategies: rice (+34.2%), shrimp (+12.3%), vegetable (+9.8%), and lotus-fish (+4.6%).

Access to public credit (CRE) marginally influenced shrimp (+0.8%) and lotus-fish (+0.6%) farming adoption but was not statistically significant for other strategies, likely due to limited credit access among surveyed households.

Finally, participation in training programs (TRA) had a strong positive impact on adoption of all strategies except vegetable production. Households participating in training were more likely to adopt salt-tolerant rice (+8.6%), vegetable production (+4.2%), and lotus-fish (+3.7%) methods.

## 4. Discussions

The findings derived from the MNL model substantiate the proposition that climate change adaptation at the household level is a complex and multidimensional process influenced by a confluence of socio-economic, demographic, institutional, and environmental factors. These results not only align with existing theoretical frameworks and empirical evidence from similar contexts in Asia and Africa but also contribute novel insights specific to Vietnam's Central Coastal region—a zone characterized by high climate vulnerability and socio-ecological diversity.

A key insight from the analysis is the gendered nature of climate adaptation. The data clearly indicate that male-headed households are more likely to adopt a broader set of adaptation strategies, including those requiring greater technical knowledge and access to resources, such as shrimp and lotus-fish farming. This pattern is consistent with prior research by Datta et al (2024), Ngigi et al (2017) and Belay et al (2017) which found that men in rural settings are often better integrated into formal and informal networks, have greater access to agricultural extension services, and possess stronger agency in household decision-making [20,24,55]. Moreover, Parry et al. (2023) and Sallah et al (2023) also emphasized that entrenched gender norms continue to limit women's access to productive assets and information, thereby constraining their adaptive capacity [17,23]. These findings reinforce the importance of integrating gender-sensitive policies in climate adaptation programs, particularly in rural agrarian settings where intra-household dynamics and social hierarchies influence resource distribution and decision-making.

The role of education as both an enabler and a filter for adaptation strategy selection is also noteworthy. While higher levels of education were associated with a reduced likelihood of engaging in labor-intensive agricultural adaptations such as shrimp or salt-tolerant rice cultivation, they were positively linked with more knowledge-intensive approaches like the lotus-fish model. This finding resonates with the work of Batunga-wayo et al (2023) and Cammarano et al (2020) who posited that education increases farmers' capacity to process information, evaluate risk, and innovate [14,61]. However, it may also suggest a movement among better-educated individuals away from traditional agriculture toward non-farm income activities, reflecting structural shifts in rural labor markets.

Age and farming experience exhibited a dual role in shaping adaptation behaviors. Older farmers were more inclined to maintain traditional practices, such as rice farming using salt-tolerant varieties, possibly due to risk aversion and accumulated knowledge. In contrast, younger or more experienced farmers appeared more open to adopting complex systems like lotus-fish farming or diversifying into aquaculture. These results parallel observations by Akum et al (2022) and Der Tambile et al (2024), suggesting that age is not only a proxy for experience but also for behavioral rigidity, whereas experience enhances capacity for strategic adaptation depending on perceived risk and opportunity [1,10].

The significance of household income in determining adaptation choices cannot be overstated. Households with higher income levels were more capable of investing in capital-intensive adaptation methods such as shrimp farming and aquaculture systems. Conversely, households with limited income were more likely to rely on traditional or subsistence strategies, and less likely to engage in innovation. The negative association between income and vegetable production may reflect both its low economic return and high labor input, discouraging wealthier households from pursuing it. This aligns with findings from Li et al (2023) and Datta et al (2024), who highlighted the pivotal role of economic resources in enabling or constraining adaptive choices, particularly in resource-scarce settings [6,20].

From an environmental perspective, both the extent of salinized land and the spatial position of agricultural plots emerged as significant determinants of strategy selection. Households experiencing severe salinity on their plots favored the adoption of more resilient rice varieties, whereas those in moderately affected areas diversified toward aquaculture or integrated farming. These patterns reflect an adaptive logic based on land suitability and are consistent with studies by Ayal and Leal (2017) and Ayal et al. (2021), who emphasized that agro-ecological conditions critically shape both the feasibility and profitability of adaptation strategies. The salinity threshold, beyond which certain crops or models become unviable, presents a strong environmental constraint that must be incorporated into local planning and land use zoning [38,63].

Institutional factors—notably access to information, credit, training, and civic networks—were also found to play a vital role in facilitating adaptation. Access to climate-related information from public authorities significantly increased the likelihood of adopting salt-tolerant rice and shrimp farming. This finding aligns with Getahun et al. (2021), who argued that timely and reliable information is essential for informed decision-making [39]. Membership in civic organizations such as the Women's Union also increased the probability of adopting all four strategies, underscoring the role of collective action and social capital in building adaptive capacity. Particularly in Vietnam, where the Women's Union has been instrumental in disseminating information, facilitating microcredit access, and organizing training sessions, these organizations serve as critical intermediaries between state-led adaptation policies and community-level action.

Interestingly, while access to public credit had a modest influence—only significantly affecting shrimp and lotus-fish adoption—it reveals deeper structural limitations in the rural credit market. Many households remain excluded from formal financial services, limiting their ability to make upfront investments in adaptive infrastructure or technologies. This finding echoes the conclusions of Saguye (2016) and Belay et al (2017), who noted that credit constraints remain one of the most significant barriers to smallholder adaptation across developing countries [41,55].

Participation in training programs had a robust and positive effect on the adoption of nearly all adaptation strategies, with the exception of vegetable production. This supports the view that climate adaptation, particularly in contexts involving technical innovations or ecological complexity (e.g., saline-affected land), requires continuous learning and institutional

support. Training enhances not only farmers' awareness but also their practical skills to implement context-appropriate practices. Similar conclusions were reached by Ngigi et al. (2017) and Asrat and Simane (2018), who emphasized the capacity-building role of agricultural extension systems and the need to institutionalize adaptive learning within rural development frameworks [24,25].

In synthesis, these findings affirm that adaptation decisions are not isolated technical choices but are deeply embedded in a broader socio-economic and institutional context. They support the literature on the socio-ecological embeddedness of adaptation behavior, highlighting that successful climate resilience planning must consider differential access to resources, power relations, gender roles, and ecological constraints. For policy, this implies a move away from "one-size-fits-all" prescriptions toward more differentiated, inclusive, and place-based adaptation strategies. Such approaches must simultaneously enhance resource access, build institutional capacity, and promote knowledge co-production between communities and decision-makers.

## 5. Conclusions and recommendations

This study has empirically examined the determinants of climate change adaptation strategies among farming households in Quang Binh province, Central Vietnam—a region acutely affected by saltwater intrusion, droughts, and other climate-induced stressors. By employing a MNL model based on a household survey of 356 respondents, the study identified a wide range of socio-economic, demographic, institutional, and environmental factors influencing households' adaptation choices. The findings confirm that climate adaptation behavior is highly context-specific and shaped by a complex interplay between resources, knowledge, and perceived environmental risks.

Notably, the results reveal that gender, education, household size, income, age, and farming experience significantly influence the likelihood of selecting specific adaptation strategies. Moreover, environmental characteristics such as the proportion and location of salinized land, institutional variables including access to climate information, credit, training, and civic networks, further explain the heterogeneity in adaptation decisions across households.

The study contributes to the growing body of literature emphasizing that adaptation to climate change is not merely a matter of access to technology or exposure to risk, but a deeply social process involving power dynamics, access to institutions, and socio-ecological knowledge. These insights are particularly relevant for transitional economies like Vietnam, where rural households are increasingly required to respond autonomously to climate threats amidst resource constraints and institutional limitations.

Based on the empirical findings and in response to the urgent need for locally tailored climate resilience strategies, this study offers a set of targeted recommendations to enhance the adaptive capacity of smallholder farmers in salinity-prone and climate-vulnerable regions of Vietnam. These recommendations are structured to reflect both the multidimensional drivers of adaptation behavior and the institutional mechanisms required for effective policy implementation.

### 5.1 Promote gender-inclusive adaptation policies

Given the significant influence of gender on adaptation behavior, policies must be designed to empower women and female-headed households. This includes improving women's access to agricultural extension services, credit programs, and training opportunities. Strengthening the role of women's unions and grassroots civic organizations will be critical for enabling inclusive participation in climate adaptation initiatives and enhancing household-level decision-making.

### 5.2 Design tailored training programs on climate-resilient agricultural practices

The study finds that participation in training programs significantly increases the likelihood of adopting technically demanding adaptation strategies, such as salt-tolerant rice cultivation and lotus-fish integrated farming. Therefore, training should move beyond awareness-raising to focus on hands-on, context-specific modules addressing soil salinity management, water use efficiency, crop diversification, and integrated aquaculture systems. These programs should be coordinated by

the Department of Agricultural Extension in partnership with local universities, research institutes, and farmer cooperatives. Training should be delivered at the commune level, with special attention to low-income and female-headed households.

### 5.3  Expand access to credit through multi-tiered rural financial systems

Access to credit was found to be a modest but significant enabler of adaptation, particularly for capital-intensive strategies. A multi-tiered approach to rural credit should be adopted by scaling up the role of the Vietnam Bank for Social Policies (VBSP), while also fostering microfinance institutions (MFIs) and agricultural credit cooperatives. Priority lending schemes should focus on households in salinity-affected regions, offering low-interest loans tied to climate-resilient agricultural investments. These credit packages should be gender-sensitive, with simplified application procedures and technical support integrated into loan programs.

### 5.4  Introduce targeted subsidies for salt-tolerant rice varieties

The widespread but income-sensitive adoption of salt-tolerant rice varieties underscores the need for targeted input subsidies. We recommend implementing a voucher-based subsidy scheme to reduce the financial burden of acquiring certified salt-tolerant seeds and associated training. The program should prioritize small and medium-scale farmers and be administered through provincial agricultural departments in coordination with commune-level authorities. Furthermore, continued public investment in research and development (R&D) of improved salt-tolerant cultivars should be linked to extension services to ensure effective dissemination and uptake.

### 5.5  Strengthen climate information systems and local advisory services

Access to timely and reliable information on climate variability, salinity levels, and adaptation techniques was found to significantly influence strategy adoption. Government agencies, in collaboration with local authorities, should invest in climate information systems that provide localized forecasts, seasonal advisories, and early warning alerts. These services should be integrated into agricultural extension channels and delivered in user-friendly formats, including SMS, radio, and village loudspeaker systems.

### 5.6  Support community-based adaptation and civic participation

Membership in local civic organizations positively influenced adaptation uptake across all strategies. Policy should encourage the development of farmer-based organizations, cooperatives, and climate adaptation clubs that facilitate peer learning, shared investment, and collective action. Programs supporting women's participation in such groups—especially through linkages with microcredit, training, and extension networks—should be prioritized as part of a broader effort to build social capital and institutional resilience.

### 5.7  Integrate land-use planning with adaptation strategies

Environmental factors such as land area, soil salinity, and land location were found to significantly shape adaptation behavior. Therefore, local land-use planning should incorporate climate vulnerability mapping and support land reallocation or conversion to more climate-compatible agricultural uses. For instance, land zoned for shrimp farming or integrated aquaculture systems should be prioritized in high-salinity zones, while rice cultivation with salt-tolerant varieties should be promoted in areas with moderate salinization.

### 5.8  Develop modular adaptation support packages

Finally, adaptation interventions should not follow a one-size-fits-all model. Instead, modular adaptation support packages—combining training, financial access, improved seed technologies, and climate information—should be designed for

different farmer profiles (e.g., land-poor households, female-headed households, experienced vs. novice farmers). These packages should be scalable, replicable, and tailored to ecological and socio-economic conditions across Vietnam's coastal regions.

## 6. Study limitations

Despite offering valuable insights into household-level adaptation to climate change in Central Vietnam, this study has several limitations. First, the cross-sectional design restricts the ability to infer causal relationships, as adaptation behaviors are inherently dynamic and may evolve over time. Second, the reliance on self-reported data introduces the possibility of recall bias and subjective interpretation, particularly in responses related to climate events and adaptation decisions. Third, the use of the Multinomial Logit (MNL) model assumes the independence of irrelevant alternatives (IIA), which may not fully reflect the real-world complexity where adaptation strategies can be interrelated or sequential. Fourth, the study focused on four prominent adaptation strategies, potentially overlooking traditional, incremental, or less visible coping practices adopted by households. Fifth, while socio-economic, institutional, and environmental variables were considered, the study did not incorporate psychological, cultural, or normative factors such as risk perception, social trust, or community beliefs, which are known to influence behavioral choices. Sixth, the geographic focus on three communes in Tuyen Hoa district limits the generalizability of the findings to other regions in Vietnam with different socio-ecological profiles. Finally, due to data constraints, the analysis could not include important variables such as market accessibility, land tenure security, migration dynamics, and remittance flows, which may significantly shape adaptation capacity and decisions. These limitations underscore the need for future research using longitudinal, mixed-method approaches and broader geographic coverage to provide a more comprehensive understanding of climate adaptation among rural households.

## Supporting information

**S1 Data.**
(XLSX)

## Author contributions

**Conceptualization:** Bui Huy Nhuong, Dinh Duc Truong, Le Huy Huan.

**Data curation:** Duong Duc Tam.

**Formal analysis:** Bui Thi Hoang Lan.

**Methodology:** Dinh Duc Truong, Le Huy Huan, Bui Thi Hoang Lan, Duong Duc Tam.

**Project administration:** Bui Huy Nhuong, Le Huy Huan, Nguyen Dieu Hang.

**Resources:** Bui Thi Hoang Lan.

**Supervision:** Le Huy Huan.

**Validation:** Le Huy Huan, Bui Thi Hoang Lan, Nguyen Dieu Hang.

**Writing – original draft:** Dinh Duc Truong.

**Writing – review & editing:** Dinh Duc Truong, Nguyen Dieu Hang, Duong Duc Tam.

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
