## [Decision Letter · Decision Letter 0]

PONE-D-24-25204Factors Influencing Farming Households’ Climate Change Adaptation Strategies in Central VietnamPLOS ONE

Dear Dr. Dinh,

Thank you for submitting your manuscript to PLOS ONE. After careful consideration, we feel that it has merit but does not fully meet PLOS ONE’s publication criteria as it currently stands. Therefore, we invite you to submit a revised version of the manuscript that addresses the points raised during the review process.

We look forward to receiving your revised manuscript.

Kind regards,

Nguyen-Thanh Son, Ph.D.

Academic Editor

PLOS ONE

Journal Requirements:

3. We note that your Data Availability Statement is currently as follows: [All data are in the manuscript and/or supporting information files] Please confirm at this time whether or not your submission contains all raw data required to replicate the results of your study. Authors must share the “minimal data set” for their submission. PLOS defines the minimal data set to consist of the data required to replicate all study findings reported in the article, as well as related metadata and methods (https://journals.plos.org/plosone/s/data-availability#loc-minimal-data-set-definition). For example, authors should submit the following data: - The values behind the means, standard deviations and other measures reported; - The values used to build graphs; - The points extracted from images for analysis. Authors do not need to submit their entire data set if only a portion of the data was used in the reported study. If your submission does not contain these data, please either upload them as Supporting Information files or deposit them to a stable, public repository and provide us with the relevant URLs, DOIs, or accession numbers. For a list of recommended repositories, please see https://journals.plos.org/plosone/s/recommended-repositories. If there are ethical or legal restrictions on sharing a de-identified data set, please explain them in detail (e.g., data contain potentially sensitive information, data are owned by a third-party organization, etc.) and who has imposed them (e.g., an ethics committee). Please also provide contact information for a data access committee, ethics committee, or other institutional body to which data requests may be sent. If data are owned by a third party, please indicate how others may request data access.

Natural Earth (public domain): http://www.naturalearthdata.com/.

Additional Editor Comments (if provided):

Reviewers' comments:

Reviewer's Responses to Questions

**Comments to the Author**

1. Is the manuscript technically sound, and do the data support the conclusions?

Reviewer #1: Yes

Reviewer #2: Partly

2. Has the statistical analysis been performed appropriately and rigorously? 

Reviewer #1: Yes

Reviewer #2: I Don't Know

3. Have the authors made all data underlying the findings in their manuscript fully available?

Reviewer #1: Yes

Reviewer #2: Yes

4. Is the manuscript presented in an intelligible fashion and written in standard English?

Reviewer #1: Yes

Reviewer #2: No

5. Review Comments to the Author

Reviewer #1: With pleasure, I reviewed the article titled: "Factors Influencing Farming Households’ Climate Change Adaptation Strategies in Central Vietnam” , ref: PONE-D-24-25204, which was submitted for publication in PLOS ONE Journal, it is both relevant and engaging. The paper was also pleasant to read.

While the paper is promising, there are some minor suggestions for improvement. If you don’t mind, I have included some comments below that I hope, will help enhance the overall quality of the manuscript. Please find them below:

*Please add a short title.

*Abstract

1. I suggest to write “most vulnerable to the impacts of climate change”

2. I think you have to start your abstract by a more general background instead of writing directly about Vietnam

3. Please consider removing the part when you specify the software you used for your statistical analyses “Data were analyzed using SPSS 23.0 software”. However, you still have to indicate the method you followed and the analyses you have done.

*Introduction

1. I really appreciated the way you started your introduction, well done. Please consider a general approach like this in your abstract.

2. Honestly, I am not really convinced by the end of your introduction, you explained the structure of the article; I don’t think it is necessary. Additionally, I think the main objective may be unclear for the readers. You explained the gap in current literature and the importance of your study. Nevertheless, it can be interesting to be clearer (eg. This study aimed at….the aim of the study was….etc.).

*Methods

1. Please add the GPS coordinates for the study area.

2. The quality of Figure 1 looks a little poor. Please consider improving it. In addition, I saw source: (2022), did the authors obtain permission from Truong for the reuse of this material? Why not create your own map in that case? It would significantly enhance the quality of this manuscript.

3. The formula of your sample size should also be numbered.

4. I think you have to separate “data collection” from “statistical analysis”

5. Please add the full name of the software you used Statistical Package for the Social Sciences (SPSS) instead of the acronym, as it was the first time it appears in the text.

*Results and discussions

1. The quality of Figure 3 is also a little poor, please consider increasing its resolution

2. Please consider adding some references from 2023 and 2024, you don’t have to limit yourself to papers related to your geographical area, you can consider adding other works to support some statements.

3. Apart from that, the discussion section was well conducted, I congratulate the authors.

*Conclusion & Implication

1. I think this part is too long, please consider shortening it

2. The part related to limitations must be put in the end of the discussion section, not in the conclusion.

Finally, I would like to thank the authors for this interesting research; I wish them good luck.

Reviewer #2: To the Authors,

I founded many written mistakes along the paper. I will mention some examples:

a. Competing interests: a T is missing when your response starts

b. '.... might reduce farming vulnerability and rikds to changes in the climate system'. Reduce (double space farming) and 'RIKDS' change to risks

c. Better sentence SYNTAX. From 'Especially, climate change is causing severe impacts on food production and economic development in the agricultural sector' TO 'Especially the food production and economic development in the agricultural sector is being impacted by Climate Change'

d. 'chains (Lien et al., 2018, Le and Nguyen, 2029).. However,'. Double full period.

e. SugGestion: replace 'quite a few' with 'several' for a formal writing.

f. 'Identified factor groups included behavioral, psychological, or socio-economic factors of the farmers.' Advice: delete the last 'factors'

g. You repeted the first factor with different understanding of the sentence: '(i), there are no studies on adaptive behavior in countries with transitional economies and strongly affected by climate change like Vietnam (Haile et al.,

2013; Dat and Truong, 2020, Mai and Truong, 2022) (ii) there are few studies on adaptive behavior in countries with transition economies and strongly affected by climate change like Vietnam, (ii)'. Otherwise, the (ii) is mentioned twice.

h. 'linked to agriculture. They are'. You are talking about AREA so you should change from 'YOU ARE' to 'It is'

i. 'the introduction introduces'. Advide: The introduction SHOWS

j. '(iii) results and discussion section presents research results on factors'. suggestion: research takeaways

k. 'September to March of the following year, the average temperature is from 22 C - 23 C'. A comma is missing after 23°C

l. last part before figure 1: 'and there is also moderate rain in December'. That was previously informed when you mentioned that between Sept - March it exists the rainy season.

m. Figure 1: The names of the regions are not easily readable.

n. Data collection and analysis: you use the word Collect many times, particularly in the 2nd paragraph.

o. 'Chau Hoa Phong Hoa and Duc Hoa Communes (Thanh, 2021).' A comma is missing to separe the regions.

p. SOmetimes you write: analySys and sometimes analyZes. Are you writing in US or EN english?

q. 'small family sizes.. The age'. DOUBLE PERIOD.

r. '42.2 % and 31.3% of people questioned said that it was more than in 2018)' the initial ( is missing.

s. 'In our study, the role of government is reflected through the institutional factors included in the model. Our results showed' I think it is better writting in 3rd person.

Also there are many misconceptions. The main: 'Natural Disaster'. Please review the term: https://unu.edu/ehs/series/5-reasons-why-disasters-are-not-natural But also:

-'Farmers' choice of adaptation strategies often depends on factors such as demographic and socio-economic characteristics': and technical/ community/ ancestral knowledge;

-'As a country with a coastline of 3,260 km, climate change and rising sea'. In that phrase you are combining 2 different concepts: You should mention all the risks (rising sea) or the global concept of climate change

-Introduction: the lack of response resources is a low adaptative capacity. It you are taking on the natural resourses please specify the Sensibility

-'drought adaptation strategy'. The adaptation strategy is FOR DROUGHT. The Drought is not an adaptation strategy.

-'The prolonged heatwave and increasing seawater intrusion'. You mentioned that the impacts were drought and seawater intrusion. Please do not consider heatwaves and drought as synonims

- saltwater intrusion, and drought are NOT Natural Disasters as you were mentioning in 'Climate change adaptation strategies of households' first paragraph.

Another thing to add, I think the full title could be Factors Influencing Farming HouseHolds´Climate Change Adaptation Strategies in Quang Binh, Central Vietnam, distinguishing it from the short one.

In the study, you were mentioning many times the awareness of Climate Change from the population but that question wasn´t included in the interview or is not formally analyzed, it appears to be an indirect conclusion. To complete with the researching information, is that you implemented the interviews in the rainy season (Nov-Dec). Do you think that is also important to make interviews all along the year? Or at least between the 2 seasons you mentioned?

Some references are missing, specifically in strong facts, such as:

Introduction: I suggest to add FAO/ UNEP bibliography in the introduction with some figures. EG: 52% of global agricultural lands are now moderately to severely degraded (Source: UNEP FI)

Introduction: which is the source to say 'the area that has suffered the most natural disasters'? It is a strong communication that should be back-up

Data Collection and analysis: 'we first selected 3 communes of Tuyen Hoa District with a high proportion of farming households and severe vulnerability to climate change'. Which is the bibliography that determine the severe vulnerability? This should be mentioned as reference because it cannot be a public perception

figure 3: one unit is missing in the representation of the 1rst graphic

6. PLOS authors have the option to publish the peer review history of their article (what does this mean? ). If published, this will include your full peer review and any attached files.

**Do you want your identity to be public for this peer review?** For information about this choice, including consent withdrawal, please see our Privacy Policy .

Reviewer #1: No

Reviewer #2: No

---

## [Author Response · Author response to Decision Letter 1]

21 Feb 2025

RESPONSE REPORT TO REVIEWERS

The authors would like to thank the editor and reviewers so much for detailed comments and suggestions. These are very valuable comments that help the authors improve the article. The authors have fully received and revised the entire article according to the comments of the reviewers. Below is the author's detailed explanation and modification (in red text) according to each comment point of the editor and reviewers. We have also reviewed and improved the entire article in terms of rationale, content, model, academic references, language and format (Please see the revised draft for more detail). The authors hope to receive considerations by the editor and reviewers.

Reviewer #1:

With pleasure, I reviewed the article titled: "Factors Influencing Farming Households’ Climate Change Adaptation Strategies in Central Vietnam, ref: PONE-D-24-25204, which was submitted for publication in PLOS ONE Journal, it is both relevant and engaging. The paper was also pleasant to read.

While the paper is promising, there are some minor suggestions for improvement. If you don’t mind, I have included some comments below that I hope, will help enhance the overall quality of the manuscript. Please find them below:

*Please add a short title.

Thanks much for the suggestion! We want this to be a case study in Vietnam and the implications of the study can be referenced for other coastal areas in Vietnam that are also affected by natural disasters in the context of climate change. Therefore, we have chosen a broader title.

*Abstract

1. I suggest to write “most vulnerable to the impacts of climate change”.

We have corrected as reviewer comment!

2. I think you have to start your abstract by a more general background instead of writing directly about Vietnam.

We have added the general background as suggested. ‘Climate change is a huge challenge to human development in the 21st century’.

3. Please consider removing the part when you specify the software you used for your statistical analyses “Data were analyzed using SPSS 23.0 software”. However, you still have to indicate the method you followed and the analyses you have done.

We have deleted ‘using SPSS 23.0 software’ and only indicate the method for data analysis.

*Introduction

1. I really appreciated the way you started your introduction, well done. Please consider a general approach like this in your abstract.

Thanks so much! We have done as you suggestion in the Abstract (please see the revised paper).

2. Honestly, I am not really convinced by the end of your introduction, you explained the structure of the article; I don’t think it is necessary. Additionally, I think the main objective may be unclear for the readers. You explained the gap in current literature and the importance of your study. Nevertheless, it can be interesting to be clearer (eg. This study aimed at….the aim of the study was….etc.).

We deleted the final paragraph about structure of the paper as comment. We also have put a sentence to clarify the objective of the study as suggestion.

‘…This study aim at analysinig the factors that influence the choice of adaptive livelihood strategies of farming households in Vietnam..’

*Methods

1. Please add the GPS coordinates for the study area.

We have added GPS coordinates for the study area Tuyen Hoa district (17.7882° N, 106.2051° E).

2. The quality of Figure 1 looks a little poor. Please consider improving it. In addition, I saw source: (2022), did the authors obtain permission from Truong for the reuse of this material? Why not create your own map in that case? It would significantly enhance the quality of this manuscript.

We have replaced the Figure with a new one with higher resolution according to your suggestion.

3. The formula of your sample size should also be numbered.

We have numbered the formula as suggested.

4. I think you have to separate “data collection” from “statistical analysis”

We include the data collection and analysis in section 2.3 (Data collection and analysis).

5. Please add the full name of the software you used Statistical Package for the Social Sciences (SPSS) instead of the acronym, as it was the first time it appears in the text.

We have added and corrected as your suggestion

*Results and discussions

1. The quality of Figure 3 is also a little poor, please consider increasing its resolution

We have replaced Figure 3 with higher resolution

2. Please consider adding some references from 2023 and 2024, you don’t have to limit yourself to papers related to your geographical area, you can consider adding other works to support some statements.

We have added some references from 2023-2024 (in the text and the lists) to clarify arguments as comment.

3. Apart from that, the discussion section was well conducted, I congratulate the authors.

Thanks so much!

*Conclusion & Implication

1. I think this part is too long, please consider shortening it. The part related to limitations must be put in the end of the discussion section, not in the conclusion.

Thank you for your comment. We have shortened the Conclusion & Implication. We have also added limitations to the discussion section as suggested.

Finally, I would like to thank the authors for this interesting research; I wish them good luck.

Once again, the authors would like to thank the reviewer so much and hope to receive the consideration from the reviewer.

Reviewer #2:

To the Authors

I founded many written mistakes along the paper. I will mention some examples:

a. Competing interests: a T is missing when your response starts.

We have checked and corrected it as suggested.

b. '.... might reduce farming vulnerability and rikds to changes in the climate system'. Reduce (double space farming) and 'RIKDS' change to risks.

We have checked and corrected it as suggested.

c. Better sentence SYNTAX. From 'Especially, climate change is causing severe impacts on food production and economic development in the agricultural sector' TO 'Especially the food production and economic development in the agricultural sector is being impacted by Climate Change'.

We have checked and corrected it as suggested.

d. 'chains (Lien et al., 2018, Le and Nguyen, 2029).. However,'. Double full period. We have checked and corrected it as suggested.

e. SugGestion: replace 'quite a few' with 'several' for a formal writing.

We have checked and corrected it as suggested.

f. 'Identified factor groups included behavioral, psychological, or socio-economic factors of the farmers.' Advice: delete the last 'factors'

We have checked and corrected it as suggested.

g. You repeted the first factor with different understanding of the sentence: '(i), there are no studies on adaptive behavior in countries with transitional economies and strongly affected by climate change like Vietnam (Haile et al., 2013; Dat and Truong, 2020, Mai and Truong, 2022) (ii) there are few studies on adaptive behavior in countries with transition economies and strongly affected by climate change like Vietnam, (ii)'. Otherwise, the (ii) is mentioned twice.

We have checked and corrected it as suggested.

h. 'linked to agriculture. They are'. You are talking about AREA so you should change from 'YOU ARE' to 'It is'

We have checked and corrected it as suggested.

i. 'the introduction introduces'. Advide: The introduction SHOWS

We have checked and corrected it as suggested.

j. '(iii) results and discussion section presents research results on factors'. suggestion: research takeaways

We have checked and corrected it as suggested.

k. 'September to March of the following year, the average temperature is from 22 C - 23 C'. A comma is missing after 23°C

We have checked and corrected it as suggested.

l. last part before figure 1: 'and there is also moderate rain in December'. That was previously informed when you mentioned that between Sept - March it exists the rainy season.

We have checked and corrected it as suggested.

m. Figure 1: The names of the regions are not easily readable. We changed the graph for more detail and clear

We have changethe grapb as suggested.

n. Data collection and analysis: you use the word Collect many times, particularly in the 2nd paragraph.

We change the word ‘collect’ for more diverse.

o. 'Chau Hoa Phong Hoa and Duc Hoa Communes (Thanh, 2021).' A comma is missing to separe the regions.

We have checked and corrected it as suggested.

p. SOmetimes you write: analySys and sometimes analyZes. Are you writing in US or EN english?

We changed all to ‘analyse’

q. 'small family sizes.. The age'. DOUBLE PERIOD.

We changed to ‘The age of the household head was used as a measure of farming experience’.

r. '42.2 % and 31.3% of people questioned said that it was more than in 2018)' the initial ( is missing.

We checked and corrected.

s. 'In our study, the role of government is reflected through the institutional factors included in the model. Our results showed' I think it is better writtin

g in 3rd person.

We have checked and corrected it as suggested.

• Also there are many misconceptions. The main: 'Natural Disaster'. Please review the term: https://unu.edu/ehs/series/5-reasons-why-disasters-are-not-natural But also:-'Farmers' choice of adaptation strategies often depends on factors such as demographic and socio-economic characteristics': and technical/ community/ ancestral knowledge;

We have checked the suggested reference and from the lituerature revew, we added following paragraph as you suggestions:

‘…This study examines the factors influencing farmers' climate variability adaptation strategies in Vietnam which mighte include:

Socio-economic factors: farm size, farm income, memberships in civic organizations, and farming experience.

Demographic factors: family size and age.

Institutional factors: access to credit services, markets, training, and climate change information….’

• 'As a country with a coastline of 3,260 km, climate change and rising sea'. In that phrase you are combining 2 different concepts: You should mention all the risks (rising sea) or the global concept of climate change.

We change to ‘As a country with a long coastline and high population density and coastal economic activities, Vietnam is considered one of the five countries most heavily affected by climate change.’

• Introduction: the lack of response resources is a low adaptative capacity. It you are taking on the natural resourses please specify the Sensibility.

We have added sensibility in the Introduction as suggested.

• -'drought adaptation strategy'. The adaptation strategy is FOR DROUGHT. The Drought is not an adaptation strategy.

We changed the sentencet to’ adaptation strategy for drought’ as comment.

• The prolonged heatwave and increasing seawater intrusion'. You mentioned that the impacts were drought and seawater intrusion. Please do not consider heatwaves and drought as synonims.

We changed it to heatway as suggested.

• Another thing to add, I think the full title could be Factors Influencing Farming HouseHolds´Climate Change Adaptation Strategies in Quang Binh, Central Vietnam, distinguishing it from the short one.

Thanks for the suggestion! We want this to be a case study in Vietnam and the implications of the study can be referenced for other coastal areas in Vietnam that are also affected by natural disasters in the context of climate change. Therefore, we have chosen a broader title.

• In the study, you were mentioning many times the awareness of Climate Change from the population but that question wasn´t included in the interview or is not formally analyzed, it appears to be an indirect conclusion

Thanks for the suggestion! We assessed people's awareness of natural disasters in the context of climate change including awareness of natural disaster trends in the last 5 years, intensity and frequency of natural disasters, and changes in the impact of natural disasters.

‘…..The study examined the awareness and adaptation strategies adopted by respondents to cope with with disasters in climate change context in local communities. The majority (85.23%) of selected households were aware of the existence of climate change and extreme events in their locality. Natural disasters frequently appeared in Tuyen Hoa in the last 5 years include floods, storms, saltwater intrusion, and drought. Frequency of occurrence of natural disasters compared to previous years according to people's objective point of view (drought, flood and saltwater intrusion), with corresponding percentages of 51.9%, 42.2 % and 31.3% of people questioned said that it was more than in 2018 while 63.4% thought that the frequency of storms was less. However, the intensity of each storm is much greater (for example, storm Wuitp appeared in October 2018) (Table 3).

As reported by households, in the study area, flood disasters often occur from August to October and last until November; drought and saltwater intrusion from April to July every year. These disasters and natural disasters affect agricultural activities, especially activities that target the cultivation of agricultural crops (rice, food crops, other crops...). Interviews said that the phenomenon of saltwater intrusion and drought is strongest from April to June and can last until July with huge damage to rice cultivation. Moreover, flooding often accompanies storms that occur in August, September, and October and can last until November (Table 3). People's awareness of climate change was quite similar to climate developments analysed in Vietnam national climate change and sea level rise scenarios (MONRE, 2020) (Figure 3).’

• To complete with the researching information, is that you implemented the interviews in the rainy season (Nov-Dec). Do you think that is also important to make interviews all along the year? Or at least between the 2 seasons you mentioned?

Due to limited resources, we only conducted the survey in one time slice - November and December (rainy season). However, because the local people have lived in the area for a long time, they have a clear experience of the impact of natural disasters in both the dry and rainy seasons. We will add this point to the research limitation section according to the reviewer's comments.

• Some references are missing, specifically in strong facts, such as:

Introduction: I suggest to add FAO/ UNEP bibliography in the introduction with some figures. EG: 52% of global agricultural lands are now moderately to severely degrade (Source: UNEP FI)

The authors have added an avove fact in the Introduction section with UNEP references as suggested by the reviewer UNEP (2016). Land restoration key to human well-being. https://www.unep.org/news-and-stories/story/land-restoration-key-human-well-being.

• Introduction: which is the source to say 'the area that has suffered the most natural disasters'? It is a strong communication that should be back-up.

Thanks for the comment; we have added references to the above sentence, as follows:

‘…This is the area that has suffered the most natural disasters and negative impacts of climate change in Vietnam over the past 20 years, with people's livelihoods closely linked to agriculture (Mai and Truong, 2022)…’

• Data Collection and analysis: 'we first selected 3 communes of Tuyen Hoa District with a high proportion of farming households and severe vulnerability to climate change'. Which is the bibliography that determine the severe vulnerability? This should be mentioned as reference because it cannot be a public perception

We add the reference source for that sentence (Quang Binh People Committee, 2022).

• Figure 3: one unit is missing in the representation of the 1rst graphic.

We have checked and correct it.

Once again, the authors would like to thank the reviewer so much and hope to receive the consideration from the reviewer.

---

## [Decision Letter · Decision Letter 1]

PONE-D-24-25204R1Factors Influencing Farming Households’ Climate Change Adaptation Strategies in Central VietnamPLOS ONE

Dear Dr. Dinh,

Thank you for submitting your manuscript to PLOS ONE. After careful consideration, we feel that it has merit but does not fully meet PLOS ONE’s publication criteria as it currently stands. Therefore, we invite you to submit a revised version of the manuscript that addresses the points raised during the review process.

We look forward to receiving your revised manuscript.

Kind regards,

Sandra Ricart, Ph.D.

Academic Editor

PLOS ONE

Journal Requirements:

Reviewers' comments:

Reviewer's Responses to Questions

**Comments to the Author**

1. If the authors have adequately addressed your comments raised in a previous round of review and you feel that this manuscript is now acceptable for publication, you may indicate that here to bypass the “Comments to the Author” section, enter your conflict of interest statement in the “Confidential to Editor” section, and submit your "Accept" recommendation.

Reviewer #1: All comments have been addressed

Reviewer #3: (No Response)

2. Is the manuscript technically sound, and do the data support the conclusions?

Reviewer #1: Yes

Reviewer #3: Yes

3. Has the statistical analysis been performed appropriately and rigorously? 

Reviewer #1: Yes

Reviewer #3: Yes

4. Have the authors made all data underlying the findings in their manuscript fully available?

Reviewer #1: Yes

Reviewer #3: Yes

5. Is the manuscript presented in an intelligible fashion and written in standard English?

Reviewer #1: Yes

Reviewer #3: Yes

6. Review Comments to the Author

Reviewer #1: Dear authors,

After reading the last version, I think that it was a good job, congratulations. Even if the revision was not very consequent, the overall level of the manuscript increased. In addition, the responses to reviewers' comments were polite, detailed and adequate. Thank you again for submitting your manuscript here and for giving me the opportunity to review it. I wish you the best.

Reviewer #3: The authors mention VIF analysis but do not present a table of VIF values. Including a supplementary table with VIF values would enhance transparency and reproducibility.

The manuscript uses “natural disaster,” a contested term in disaster studies. While the authors cite UNU-EHS to justify their use, it’s recommended to use “climate-related hazards” or “weather extremes” for conceptual accuracy.

Although much improved, minor grammatical issues remain (e.g., “trainning” should be “training”; “rikds” should be “risks” – though corrected in some parts, these still appear occasionally).

The authors have added 2023–2024 references, which is appreciated. However, more diversity in geographic scope (i.e., more studies from other vulnerable countries in Asia or Africa) would further strengthen the literature review.

The updated version of Figure 3 still lacks units on the first graph axis.

Source: Study results (2024) ( Not necessary to add this – if this is your results.

Please use the Vancouver referencing style.

Include continuous line numbering for the text.

Conclusions- Tie the findings more clearly to policy — e.g., what type of training interventions are needed? Who should provide credit (state, microfinance)? Should salt-tolerant rice be subsidised?

7. PLOS authors have the option to publish the peer review history of their article (what does this mean? ). If published, this will include your full peer review and any attached files.

**Do you want your identity to be public for this peer review?** For information about this choice, including consent withdrawal, please see our Privacy Policy .

Reviewer #1: No

Reviewer #3: No

---

## [Author Response · Author response to Decision Letter 2]

11 Jun 2025

RESPONSE REPORT TO REVIEWER

The authors would like to thank the reviewer so much for detailed comments and suggestions. These are very valuable comments that help the authors improve the article. The authors have fully received and revised the entire article according to the comments of the reviewer. Below is the author's detailed explanation and modification (in red text) according to each comment point of the editor and reviewers. We have also reviewed and improved the entire article in terms of rationale, novelties, research questions, academic contributions, references, discussions, language and format (Please see the revised paper for more detail). The authors hope to receive considerations by the editor and reviewer.

Reviewer #3:

Comment 1:

The authors mention VIF analysis but do not present a table of VIF values. Including a supplementary table with VIF values would enhance transparency and reproducibility.

Response 1:

Thank you so much for your valuable comment. The authors have added the VIF column in Table 3. Accordingly, the VIF values are all less than 2, so there is no multicollinearity in the analysis model.

Comment 2:

The manuscript uses “natural disaster,” a contested term in disaster studies. While the authors cite UNU-EHS to justify their use, it’s recommended to use “climate-related hazards” or “weather extremes” for conceptual accuracy.

Response 2:

Thank you for your valuable scientific comment. We have replaced ‘natural disasters’ with ‘climate-related hazards’ in the article as per your comment.

Comment 3:

Although much improved, minor grammatical issues remain (e.g., “trainning” should be “training”; “rikds” should be “risks” – though corrected in some parts, these still appear occasionally).

Response 3:

Thank you for your comment. We have rewritten many of the main contents of the revised paper in English academic style, and have checked and corrected all spelling and typos in the paper according to your comment.

Comment 4:

The authors have added 2023–2024 references, which is appreciated. However, more diversity in geographic scope (i.e., more studies from other vulnerable countries in Asia or Africa) would further strengthen the literature review.

Response 4:

Thank you for this important comment. The authors have added more recent references in Asia and Africa as suggested. Some examples include:

1. Kone S, Balde A, Zahonogo P, et al. A systematic review of recent estimations of climate change impact on agriculture and adaptation strategies perspectives in Africa. Mitig Adapt Strateg Glob Change. 2024;29:18.

2. Adam M, MacCarthy DS, Traoré PCS, et al. Which is more important to sorghum production systems in the Sudano Sahelian zone of West Africa: climate change or improved management practices? Agric Syst. 2020;185:102920.

3. Alemseged A, Getahun YS, Belete Y, et al. A systematic literature review on factors influencing climate-smart agriculture adoption among African farmers. Front Environ Econ Rev. 2023;1:1356335.

4. Cammarano D, Valdivia RO, Beletse YG, et al. Integrated assessment of climate change impacts on crop productivity and income of commercial maize farms in northeast South Africa. Food Sec. 2020;12:659–678.

5. Akum RA, Atiah WA, Amekudzi LK, et al. Climate variability and impacts on maize yield in Ghana: a systematic review. Q J R Meteorol Soc. 2022;148(742):185–198.

6. Batunga wayo P, Habarugira V, Vanclooster M, et al. Confronting climate change and livelihood: smallholder farmers’ perceptions and adaptation strategies in northeastern Burundi—a systematic review. Reg Environ Change. 2023;23(1):47.

7. Der Tambile E, Ramachandran VS, Rajendrakumar S, et al. Rural livelihoods sustainability in South Asia and Africa: a systematic review with bibliometric analysis. Environ Sustain Indic. 2024;

8. Sallah M, Antoniades G, Kourkafas GE, et al. Enabling climate change adaptation in coastal systems: a bibliometric review of empirical studies. Earth’s Future. 2023;1–22.

9. Datta P, Behera B, Rahut DB. Climate change and Indian agriculture: a systematic review of farmers’ perception, adaptation, and transformation. Environ Dev Sustain. 2024;26(3):4123–4142.

10. Sallah M, Antoniades G, Kourkafas GE, Tsiouri M. Enabling climate change adaptation in coastal systems: a bibliometric review of empirical studies. Earth’s Future. 2023;11(10):e2023EF004912.

Comment 5:

The updated version of Figure 3 still lacks units on the first graph axis.

Response 5

The authors have checked and added according to the comment in Figure 3.

Comment 6:

Please use the Vancouver referencing style. Include continuous line numbering for the text.

Response 6:

Thank you so much. We have adjusted and used the Vancouver referencing style in the revised paper. The authors have also added line numbering for the text as suggested.

Comment 7:

Conclusions- Tie the findings more clearly to policy — e.g., what type of training interventions are needed? Who should provide credit (state, microfinance)? Should salt-tolerant rice be subsidised?

Response 6:

We thank the reviewer for this insightful comment. In response, we have revised the Recommendations section to explicitly link the empirical findings to specific policy actions and institutional responsibilities. In particular, we now detail the types of training interventions needed, identify key stakeholders for credit provision, and discuss the potential role of targeted subsidies for salt-tolerant rice. These additions aim to improve the practical relevance and policy utility of the study’s conclusions. For example:

Design Tailored Training Programs Focused on Climate-Resilient Agricultural Practices

Based on the significant role of training participation in promoting adaptation behaviors, especially for technically demanding strategies such as salt-tolerant rice cultivation and lotus-fish integration, it is recommended that training programs be customized to meet local needs. These should go beyond general awareness-raising and focus on hands-on, demonstration-based modules covering soil salinity management, integrated farming techniques, and irrigation scheduling. Training should be organized regularly at the commune or village level and facilitated through a partnership between the Department of Agricultural Extension, local universities, and farmer cooperatives.

Expand Access to Credit through Multi-Tiered Financial Systems

Given the finding that access to credit modestly improves adoption of capital-intensive strategies such as shrimp and lotus-fish farming, policy should promote the development of a multi-tiered rural finance system. This could include expanding the outreach of the Vietnam Bank for Social Policies (VBSP) in parallel with mobilizing microfinance institutions (MFIs) and credit unions. Priority should be given to low-interest, climate-resilient loans targeting smallholders in salinity-prone regions, especially female-headed households or members of local civic organizations.

Introduce Targeted Subsidies for Salt-Tolerant Rice Varieties

As salt-tolerant rice varieties were among the most widely adopted strategies but still showed sensitivity to land characteristics and income levels, we recommend the introduction of targeted input subsidies. These should focus on reducing the cost burden of seed acquisition and initial training for poor and medium-income households. The subsidies could be implemented through voucher-based schemes coordinated by provincial agricultural departments, with monitoring mechanisms to ensure proper targeting and transparency. In parallel, public investment in research and development (R&D) of high-yield, climate-resilient rice strains should be continued and linked to extension services.

Once again, the authors would like to thank the reviewer so much and hope to receive the consideration and acceptance from the reviewer!

---

## [Editor Report · Decision Letter 2]

Factors Influencing Farming Households’ Climate Change Adaptation Strategies in Central Vietnam

PONE-D-24-25204R2

Dear Dr. Dinh,

We’re pleased to inform you that your manuscript has been judged scientifically suitable for publication and will be formally accepted for publication once it meets all outstanding technical requirements.

Kind regards,

Sandra Ricart, Ph.D.

Academic Editor

PLOS ONE

---

## [Editor Report · Acceptance letter]

PONE-D-24-25204R2

PLOS ONE

Dear Dr. Truong,

I'm pleased to inform you that your manuscript has been deemed suitable for publication in PLOS ONE. Congratulations! Your manuscript is now being handed over to our production team.

Kind regards,

on behalf of

Dr. Sandra Ricart

Academic Editor

PLOS ONE